# Theory-Level Autoformalization:
# From Isolated Statements to Unified Formal Knowledge Bases

**Marcus J. Min** [1]  **Mike He** [2]  **Zhaoyu Li** [3]  **Zixuan Yi** [1]  **Sharad Malik** [2]  **Aarti Gupta** [2]  **Xujie Si** [3]  **Osbert Bastani** [1]

## Abstract

Autoformalization translates informal natural language into formal, machine-verifiable languages. While most work focuses on individual statements, real formalization efforts are inherently theory-level: they require an entire web of axioms, definitions, and lemmas before target theorems can even be stated. In this position paper, we argue for theory-level autoformalization: formalizing complete theories, including all their inter-dependencies, as structured libraries. We examine the significance of this shift, address alternative views, identify open challenges, and propose three promising paths forward. Our survey of autoformalization is available at https://github.com/marcusm117/Awesome-Autoformalization.

## 1. Introduction

We define **Theory-Level Autoformalization** as the task of automatically formalizing the entire theoretical context within a given scope, including axioms, definitions, notations, examples, lemmas, theorems, proofs, tactics, and all their inter-dependencies, as a coherent formal library.

As depicted in the **"Theory-Level Autoformalization Tower"** (Figure 2), formalizing even a single target theorem like the Pythagorean theorem requires first constructing multiple layers of axiomatic primitives, derivative definitions, and proof infrastructure. This contrasts with statement-level autoformalization, which translates individual theorems in isolation and implicitly relies on formal theoretical contexts like Lean Mathlib (Mathlib Community, 2020). For do-

---
[1]University of Pennsylvania [2]Princeton University [3]University of Toronto. Correspondence to: Marcus J. Min <marcmin@seas.upenn.edu>, Mike He <mikehe@princeton.edu>, Sharad Malik <sharad@princeton.edu>, Aarti Gupta <aartig@cs.princeton.edu>, Xujie Si <six@cs.toronto.edu>, Osbert Bastani <obastani@seas.upenn.edu>.

*Proceedings of the 43rd International Conference on Machine Learning*, Seoul, South Korea. PMLR 306, 2026. Copyright 2026 by the author(s).

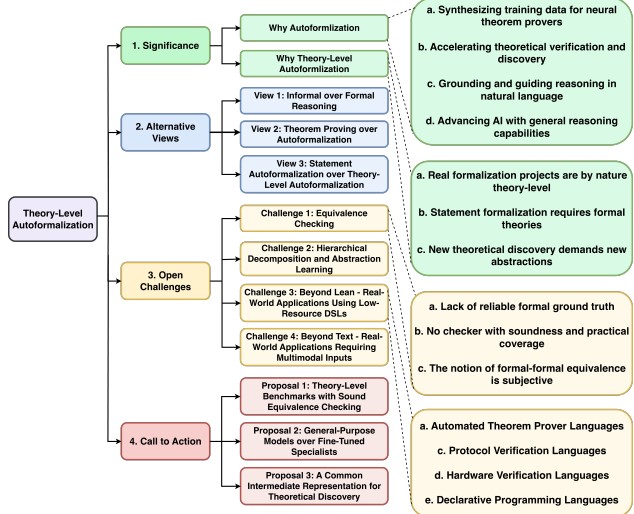

*Figure 1.* Overview of this position paper. We argue for theory-level autoformalization by examining its significance (Section 2), addressing alternative views (Section 3), identifying open challenges (Section 4), and proposing paths forward (Section 5).

mains not yet supported by such libraries, the main work is precisely to construct those missing theoretical contexts.

The need for this shift is evident from real-world formalization efforts (Table 1). The Kepler conjecture is a single statement, yet its formalization required 11 years of constructing an entire web of definitions and supporting lemmas (Hales et al., 2015). AI-assisted approaches can largely accelerate such efforts: the human formalization of the prime number theorem took **1.5 years** (Avigad et al., 2006), while the AI-assisted formalization of its quantitative refinement took only **3 weeks** (Math Inc., 2025a). Theory-level autoformalization is necessary as statements implicitly require that the surrounding theories are already formalized. Most importantly, transformative new results hinge on new abstractions that reorganize and compress our entire knowledge base, enabling proofs that could not even be formulated.

As mapped out in Figure 1, we examine the significance of theory-level autoformalization (Section 2) and address alternative views (Section 3). We then identify open challenges in evaluation, decomposition, low-resource domain-specific languages, and multimodal inputs (Section 4), and propose paths forward (Section 5).

## 2. Significance: Why Do We Need Theory-Level Autoformalization?

### 2.1. Why Autoformalization?

**a. Synthesizing training data for neural theorem provers.** Progress in neural theorem proving is tightly dependent on the availability of large, high-quality formal corpora (Xin et al., 2024; Lin et al., 2025a). Autoformalization is the most scalable way to synthesize high-quality informal–formal parallel data from the vast supply of informal statements and proofs. It addresses two complementary bottlenecks: (i) *Statement Autoformalization* turns natural-language theorems and conjectures into formal declarations, expanding coverage of interesting or challenging statements that are not yet formalized; and (ii) *Proof Autoformalization* turns informal proofs into machine-checkable proof scripts, directly producing high-quality data for theorem proving.

**b. Accelerating theoretical verification and discovery.** Formalization of important theorems and crucial engineering systems not only enables identifying potential mistakes, but also facilitates certified extensions and future theoretical discoveries. Table 1 surveys representative formalization efforts across 4 domains: mathematics, science, software, and hardware. The common bottleneck is that these projects take years, some even decades (Leroy, 2009) of collaborative effort by a team of domain experts. Autoformalization thus becomes a promising way to accelerate such processes.

**c. Grounding and guiding reasoning in natural language.** Natural-language reasoning by LLMs is prone to hallucination and factual inconsistency (Ji et al., 2023). Autoformalization improves reliability by translating informal arguments into formally checkable representations (Yang et al., 2022; Zhou et al., 2024), which provides two complementary benefits: (i) *Grounding* rules out incorrect steps by finding contradictions, ill-typed definitions, or non-existing premises. (ii) *Guiding* provides checker feedback as a guidance signal, enabling self-refinement loops. Autoformalization also benefits human informal reasoning, where arguments often omit routine steps, gloss over delicate cases, or carry unnecessary assumptions (Autexier & Fiedler, 2006). Autoformalization helps human researchers identify the handwavy steps and redundant premises, leading to more rigorous and elegant arguments. Furthermore, autoformalization can lower the barrier to using proof assistants by letting domain experts work closer to their natural mathematical language, while the autoformalizer handles much of the formal syntax and bookkeeping (Shulman, 2024).

**d. Advancing AI with general reasoning capabilities.** Szegedy (2020) envisions a general reasoning AI system powered by autoformalization: the autoformalizer generates candidate formalizations; the reasoner leverages formal verifiers to keep only correctly proved statements and

then checks the semantic alignment with the informal input. Empirically, model performance correlates strongly across mathematics, programming, and other reasoning benchmarks (Emberson & Edelman, 2026), suggesting a shared underlying capability. Training-wise, Pang et al. (2025) demonstrates that a math-only initial RL stage with verifiable rewards can elicit reasoning behaviors that transfer across domains without specialized reward models, yielding consistent multi-domain gains.

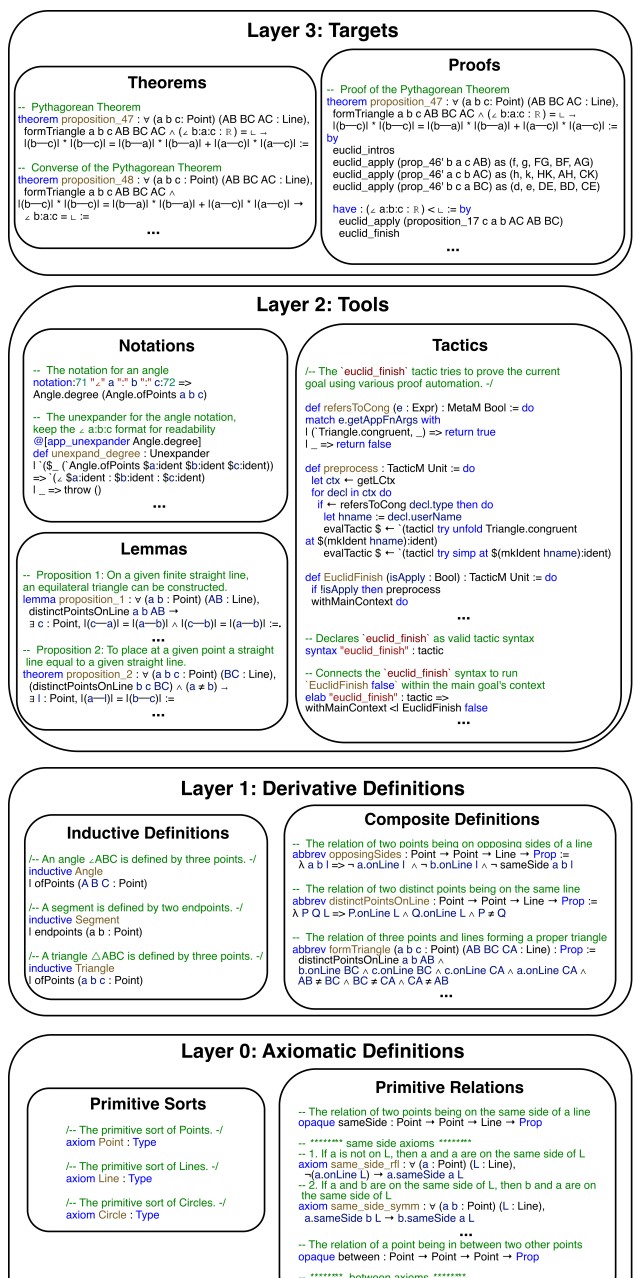

*Figure 2.* The **"Theory-Level Autoformalization Tower"** illustrated with the formal SystemE (Avigad et al., 2009) of Euclidean geometry implemented by Murphy et al. (2024) in Lean.

*Table 1.* Representative formalization projects across domains, which require substantial time and effort from human experts.

| Domain | Formalization Project | Verification Tool | Start | Duration |
|---|---|---|---|---|
| Mathematics | Four Color Theorem (Gonthier, 2007) | Coq[1] (Barras et al., 1997) | 2000 | 5 years |
| | Kepler Conjecture (Hales et al., 2015) | HOL Light[2] (Harrison, 2025) | 2003 | 11 years |
| | Odd Order Theorem (Gonthier et al., 2013) | Coq | 2006 | 6 years |
| | Liquid Tensor Experiment (Commelin et al., 2022) | Lean | 2020 | 1.5 years |
| Science | Chemical Physics (Bobbin et al., 2023) | Lean | 2022 | 1 year |
| | Applied PDEs (Deniz, 2024) | HOL Light | 2022 | Ongoing |
| Software | CompCert (Leroy, 2009) | Coq | 2005 | Ongoing |
| | CertiKOS (Shao & Ford, 2010) | Coq | 2010 | Ongoing |
| | Vellvm (Zhao et al., 2012) | Coq | 2012 | Ongoing |
| Hardware | ISA-Formal (Reid et al., 2016) | Verilog model checkers | 2011 | 5 years |
| | CORE-V-Verif (OpenHW Group, 2019) | UVM (IEEE, 2020) | 2019 | Ongoing |

[1]Coq has been renamed to Rocq since 2025, see `https://rocq-prover.org/`
[2]HOL Light was the main proof assistant; Isabelle (Nipkow et al., 2002) was also used for the classification of tame graphs.

## 2.2. Why Theory-Level Autoformalization?

As depicted in the **"Theory-Level Autoformalization Tower"** (Figure 2), to formalize our target theorems with proofs, we must start from the axiomatic definitions in **Layer 0**: primitive sorts like `Points` and primitive relations like `sameSide` governed by axioms. Based on the primitive definitions, we can construct more complex derivative definitions in **Layer 1** that involve inductive types like `Angle` and composite relations such as `formTriangle` that combine multiple primitive notions. On top of the definitions, **Layer 2** provides the tooling infrastructure: notations that make formal statements readable; lemmas and tactics that facilitate subsequent proofs. Only with all these layers in place can we state and prove the target theorems in **Layer 3**. The best current method achieves 71.4% on statements in **Layer 3** (Min et al., 2026), assuming **Layers 0–2** already formalized by humans. Yet, no current method even attempts to autoformalize these lower layers.

Theory-level autoformalization is, therefore, a systems engineering endeavor to construct a coherent library of formal components that hinge on each other. It is necessary and imminent for the following 3 reasons:

**a. Real formalization projects are by nature theory-level.**
The real-world formalization efforts surveyed in Section 2.1, from the Four Color Theorem to CertiKOS, are not isolated statement translations but large-scale formal theory constructions. The Kepler conjecture is indeed a single statement, but it cost 11 years to formalize the entire web of definitions and supporting lemmas from scratch (Hales et al., 2015). Similarly, the Liquid Tensor Experiment required formalizing substantial portions of condensed mathematics before the target theorem could even be stated (Commelin et al., 2022). Theory-level autoformalization is thus necessary to reduce the dominant cost of expert labor.

**b. Statement formalization requires formal theories.**
Many existing statement autoformalization tasks appear tractable because the heavy lifting is implicitly done by a mature formal environment of proof assistants and standard libraries. Modern proof assistants such as Lean (Moura & Ullrich, 2021) provide expressive foundations and powerful metaprogramming support, and large human-formalized standard libraries like Mathlib (Mathlib Community, 2020) already supply a vast collection of definitions, notation, and lemmas that can be reused directly. If we want to formalize statements in areas not well-supported by Mathlib, for example, numerical analysis, then the main work is precisely to construct the missing formal theoretical context. Even for standard undergraduate mathematics, there are still missing sub-theories in Lean Mathlib (Lean Community, 2026). Therefore, extending statement autoformalization to new domains necessitates theory-level capabilities.

**c. New theoretical discovery demands new abstractions.**
Significant theoretical discoveries i.e. proofs of difficult theorems typically require new abstractions that refactor and compress our representation of knowledge, enabling arguments that could not even be stated before. Abstract algebra introduced groups, rings, and fields that abstract away specific structures like the integers or polynomials, focusing on analyzing symmetries. This shift led to Galois's proof that the general quintic cannot be solved by radicals, a result that depends on recognizing that the symmetric group $S_5$ is not solvable. Category theory abstracted away internal structure to focus on morphisms between objects. Grothendieck's invention of étale cohomology (Milne, 2013)—defining a richer cohomology theory for algebraic varieties over finite fields by replacing the category of open sets with the category of étale mappings—ultimately enabled Deligne's proof of the Weil conjectures (Deligne, 1980), resolving the number theory problem in a geometric way. These examples

illustrate that theoretical progress hinges on the creation of new abstractions that compress knowledge and unlock more general results. After theory-level autoformalization from all of mathematics, science, and engineering, we will have an enormous codebase of formalized knowledge. Refactoring this codebase allows us to identify common structures and patterns across various domains, abstract them out, compress our knowledge base, and eventually use these new abstractions to prove new results.

To conclude, theory-level autoformalization is an inevitable step toward the vision of an AI system that can automate real-world verification and genuine theoretical discovery.

## 3. Alternative Views: Why Not Otherwise?

### 3.1. View 1: Informal over Formal Reasoning.

Informal natural-language reasoning has made remarkable progress since the release of OpenAI o1 (OpenAI, 2024) and DeepSeek-R1 (Guo et al., 2025). The IMO 2025 results (Luong & Lockhart, 2025; Wei, 2025) further demonstrate that informal reasoning alone can tackle difficult competition problems without formal verification. Practically, informal reasoning is also more flexible: it requires no specialized syntax and is not restricted to domains, thus benefiting from massive training data compared to formal reasoning.

**Counterargument.** First, as discussed in Section 2.1, formal methods ground and guide informal reasoning for both AI and humans by catching errors and providing verified feedback. Second, formal proofs enable modular trustworthiness: proofs of difficult theorems are hard to read regardless of whether they are written in natural or formal language, but formal proofs can be mechanically checked, allowing collaborators to trust each other's contributions without inspecting every detail. This empowers large-scale collaboration among human experts and AI systems. Third, informal reasoning scales solely through learning from data, which may eventually plateau. In contrast, formal reasoning can scale through both **learning** and **search**—the only two methods that Sutton (2019) identifies to scale reliably with compute. Formal reasoning is thus complementary to informal reasoning with new results found via search.

### 3.2. View 2: Theorem Proving over Autoformalization.

Since the goal is to prove and discover new theorems, theorem proving, especially the generation of proof scripts and verification conditions, directly addresses this objective. From this perspective, autoformalization appears secondary and is merely one of several ways to synthesize training data for neural-based theorem proving.

**Counterargument.** As analyzed in Section 2.1, autoformalization has value that extends well beyond training data

synthesis: it accelerates verification and discovery, grounds natural-language reasoning, and advances general AI reasoning capabilities. This is especially evident in hardware verification, where the real bottleneck is formalizing specifications rather than proving properties, as the latter is typically handled by mature model checking tools. Even in the context of interactive theorem proving, the prover presupposes a formal statement as the goal, but where does this statement come from? We do not care about proving arbitrary formal statements, but only the ones that matter to human mathematicians or have significant downstream impact like the correctness of compilers and operating systems. Autoformalization is what produces these statements in the first place, making it the fuel of theorem proving. Moreover, even when both tasks target proofs, they differ in objective: theorem proving seeks *any* valid proof, while proof autoformalization faithfully translates a *specific* informal argument (see Appendix B for detailed discussions).

### 3.3. View 3: Statement Autoformalization over Theory-Level Autoformalization.

Statement autoformalization is so far a more tractable problem with various benchmarks and evaluation methods. Recent human-AI collaborations (Math Inc., 2025a;b; 2026) demonstrate that complicated theorems like the strong prime number theorem can be autoformalized with the following pipeline: human experts first provide a "blueprint"—a well-decomposed dependency graph of lemmas and definitions—and an LLM-based agent follows this blueprint to autoformalize each lemma in topological order, essentially performing a sequence of statement-level translations. The agent then proves the lemmas, optionally guided by informal proofs. Given these positive signals, one might argue that statement-level methods should be prioritized.

**Counterargument.** First, as explained in Section 2.2, statement autoformalization implicitly depends on theory-level autoformalization. The strong prime number theorem formalization by Math Inc. (2025a) was mostly built upon Lean Mathlib, which already contains the foundational theories of complex analysis, harmonic analysis, and number theory; without these formal foundations, the theorem cannot even be stated. For domains not yet supported by mature libraries, theory-level autoformalization is the only path forward. Second, even for well-supported domains, the human-blueprint approach does not scale. These blueprints are not natural textbook prose, but fine-grained decompositions into minimal easy-to-prove lemmas, written in a style closer to formal language than to informal mathematical exposition. Creating such blueprints still requires substantial time and effort from human experts. Finally, as argued in Section 2.2, theory-level autoformalization enables the discovery of new mathematical abstractions, which are essential tools for tackling the most challenging open conjectures.

# 4. Open Challenges: What's Missing for Theory-Level Autoformalization

## 4.1. Challenge 1: Equivalence Checking

Equivalence checking is the core of progress in autoformalization: it enables meaningful benchmarking and provides reward signals for training. Yet unlike theorem proving, where the underlying solver or type checker delivers definitive verdicts, autoformalization not only lacks reliable oracles, but even the very notion of equivalence is subjective.

**a. Lack of reliable formal ground truth.** Since there is no sound automated way to directly check semantic equivalence between a generated formalization and its informal input, reliable evaluation requires comparing against human-written formal ground truth. However, existing statement autoformalization benchmarks contain substantial errors. For example, Poiroux et al. (2025) identified and corrected 118 human formalization mistakes in ProofNet (Azerbayev et al., 2023), a 31.8% error rate among its 371 problems. For another, at least 58 errors have been found and fixed in PutnamBench (Tsoukalas et al., 2024) since its publication in November 2024, an 8.6% error rate among 672 Lean formalizations. For definition formalization, the only dedicated benchmark is proposed by Zhang et al. (2025a), consisting of just 56 definitions from Wikipedia and 30 from arXiv papers. For proof autoformalization, the only dedicated benchmark and metric is ProofFlowBench with Proof-Score (Cabral et al., 2025), containing 184 undergraduate-level statements with proofs. For theory-level autoformalization, no benchmarks currently exist yet.

**b. No checker with soundness and practical coverage.** Even with reliable ground truth, there exists no checker that is both sound and has a high coverage of non-trivial equivalence cases. Most autoformalization evaluations rely on LLM-as-a-judge: backtranslating the formalization to informal and comparing it with the input informal using LLMs (Ying et al., 2025; Gao et al., 2025), training embedding models to predict similarity scores (Lu et al., 2024), or building LLM-based scorer agents that evaluate component-wise consistency (Xuejun et al., 2025; Wang et al., 2025b). These methods provide no soundness guarantee.

Sound alternatives exist but are overly restrictive. Exact match cannot handle semantically equivalent transformations: $\forall n : \mathbb{N}, P(n)$ and $\neg\exists n : \mathbb{N}, \neg P(n)$ are logically equivalent but syntactically different. Logical equivalence handles such cases but fails when background theory knowledge is required: `rectangle(a) ∧ rhombus(a)` and `square(a)` are equivalent in Euclidean geometry but not logically equivalent, since they can be interpreted as arbitrary relations in other models. Definitional equivalence in proof assistants like Lean addresses this by unfolding definitions via $\delta$-reduction (Bailey, 2025), but still cannot equate

`m + 0` with `0 + m`. This is because `Nat.add` is defined by recursion on the second argument: `m + 0` reduces to `m` immediately, but `0 + m` gets stuck on the abstract variable `m`. Relating them requires propositional rewriting with lemmas like `Nat.add_comm`, proved by induction.

However, unrestricted propositional equivalence is not sound: any two true statements in the background theory $T$ become equivalent, making $1 + 1 = 2$ equivalent to Fermat's Last Theorem. The solution is to restrict the background theory to an admissible fragment. Liu et al. (2025) proposed "Extended Definitional Equivalence," but their implementation lacks soundness because an LLM generates the equivalence proof, with restrictions only on allowed tactics rather than the background theory itself. Poiroux et al. (2025) implemented a deterministic checker (BEq+) that forbids global context and proof search, permitting only computational automation tactics like `simp_all_arith!`, and `noncomm_ring`. This achieves 98.0% precision and 48.3% recall, compared to 100% precision and 30.9% recall for pure definitional equivalence. However, false positives remain possible when both statements are independently easy-to-prove, such as `n * 1 = n` and `n + 0 = n`: since both are provably true by these tactics, the biconditional `n * 1 = n ↔ n + 0 = n` can be proved despite them being fundamentally different properties.

**c. The notion of formal-formal equivalence is subjective.** The fundamental bottleneck that has been overlooked in previous autoformalization literature is that even for two formal statements, equivalence is inherently subjective. What counts as "equivalent" depends on how much implicit computation the evaluator performs automatically.

Consider a ground truth statement $\int_0^1 2x^3 \ln(x^2+1)\,dx > 0$ and a prediction $\int_0^1 2x^3 \ln(1+x^2)\,dx > 0$. To most readers, these appear identical because applying commutativity of addition is so effortless that it happens unconsciously. But what if the prediction is instead $\frac{1}{4} > 0$? Most would judge this as inequivalent, yet the integral does evaluate to exactly $\frac{1}{4}$. For an expert who can perform this integration mentally, the two statements may seem just as obviously equivalent. This subjectivity becomes even more apparent in boundary cases. The statement $\int_0^1 2x^3 \ln[(x-1)^2 + 2x]\,dx > 0$ is mathematically equivalent to the ground truth because $(x-1)^2 + 2x = x^2 + 1$. To someone fluent in algebraic manipulation, recognizing this identity is immediate; to others, it requires explicit expansion and simplification.

The perceived "degree of equivalence" thus varies with the evaluator's background knowledge and computational fluency. Any equivalence checker must choose a threshold for how much computation and reasoning to permit, and different such thresholds yield different judgments about what constitutes a correct formalization.

### 4.2. Challenge 2: Hierarchical Decomposition and Abstraction Learning

As discussed in Section 3.3, recent human-AI collaborations (Math Inc., 2025a;b; 2026) have formalized substantial theorems by following human-written blueprints. We consider these efforts **"Semi-Theory-Level Autoformalization"** because 2 critical gaps remain: (i) *Hierarchical Decomposition:* blueprints are fine-grained dependency graphs of minimal lemmas, not natural prose—creating them still requires significant expert effort. (ii) *Abstraction Learning:* these projects rely on Mathlib for definitions; for domains without mature libraries, definition autoformalization remains unexplored and underrepresented in training data.

**a. Hierarchical Decomposition.** In neural theorem proving, decomposition has proven effective for breaking proof goals into manageable subgoals. Early work explored different decomposition strategies: LEGO-Prover (Wang et al., 2023) builds a growing library of reusable lemmas during proof search, Zhao et al. (2023) use diffusion models to select subgoal demonstrations, and POETRY (Wang et al., 2024a) outlines proof structure with placeholders before recursively filling in subgoals. The latest systems have scaled these ideas successfully. DeepSeek-Prover-V2 (Ren et al., 2025) uses reinforcement learning to train LLMs to do decomposition, achieving 88.9% on miniF2F (Zheng et al., 2022). BFS-Prover-V2 (Xin et al., 2025) uses a planner agent for decomposition with multiple prover agents collaborating via a shared proof cache, reaching 95.1% on miniF2F. Hilbert (Varambally et al., 2025) combines a specialized prover with a general reasoning model, recursively decomposing into subgoals when both direct attempts fail, and tries them on the new subgoals until success.

For statement autoformalization, decomposition has only begun to receive attention since 2025. Xuejun et al. (2025) first use decomposition for evaluation: an LLM judge decomposes both the informal statement and the predicted formalization into premises and conclusions, then assesses their semantic alignment. DRIFT (Zhang et al., 2025b) decomposes informal statements into sub-queries, each pairing a natural language phrase with a predicted formal representation to guide dependency retrieval from Mathlib. Aria (Wang et al., 2025b) decomposes the informal statement into a dependency graph of concepts, and formalizes each node bottom-up from leaf nodes grounded in Mathlib. DNA (Min et al., 2026) recursively decomposes the informal statement into a nested structure of quantifiers, premises, and conclusions until each leaf is an atomic proposition, then translates and composes the formal outputs.

However, these methods remain insufficient for theory-level blueprint generation. The problems they handle are only high school mathematics with shallow dependency graphs, and they decompose only one proof or statement at a time.

Blueprint generation, by contrast, requires decomposing entire textbooks into coherent dependency graphs with dozens of major theorems and deeply intertwined dependencies.

**b. Abstraction Learning.** The goal is to discover reusable patterns from data that can be applied to new tasks. In program synthesis, library learning identifies common subroutines across program corpora (Ellis et al., 2021; Wang et al., 2024b). In theorem proving, systems build libraries of lemmas that streamline future proofs (Zhou et al., 2022; Wang et al., 2023). In LLM reasoning, tool learning allows models to construct reusable utilities for specific domains (Yuan et al., 2024; Qu et al., 2025). In autoformalization, the goal is to curate libraries of reusable mathematical definitions. Patel et al. (2024) first proposed the concept of definition formalization so that they can be reused when formalizing theorems, and Zhang et al. (2024) first experimented with the autoformalization of definitions extracted from IsarMathLib (Kolodynski, 2019). Def_Wiki and Def_ArXiv (Zhang et al., 2025a) are the only dedicated definition formalization benchmarks so far with 56 and 30 definitions respectively.

DNA (Min et al., 2026) is the first to apply abstraction learning to statement autoformalization by extracting common concepts across a corpus, constructing a dependency graph of these concepts, and formalizing them as reusable definitions. Nonetheless, this framework handles only composite definitions of mathematical relations, which are built upon other lower-level definitions already present in the library. No existing work addresses axiomatic definitions or algorithmic definitions. In general, both the scale and the scope of current abstraction learning benchmarks and methods fall well short of what theory-level autoformalization demands.

Abstraction learning serves 2 roles in theory-level autoformalization. (i) *Definition Formalization*: extracting and formalizing reusable definitions directly tackles derivative definitions in the Theory-Level Autoformalization Tower (Figure 2). (ii) *Knowledge Compression*: as discussed in Section 2.2, significant discoveries rely on inventing new abstractions that refactor existing knowledge into more compact representations, enabling results that could not be stated before. The 2 roles correspond to complementary methodologies: (i) *Agent-driven*, where an LLM summarizes recurring concepts from an informal corpus (Min et al., 2026), are well-suited for definition formalization since no formal corpus yet exists for symbolic methods to operate on, and mathematical concepts in natural language are already highly abstracted. (ii) *Symbolic* (Ellis et al., 2021; Zhou et al., 2022), which identify common structures by analyzing a formal corpus directly, are a natural fit for knowledge compression once a formal library is available: they make the library more compact and modular, revealing hidden connections between seemingly distant domains.

### 4.3. Challenge 3: Beyond Lean - Real-World Applications Using Low-Resource DSLs

Real-world applications of autoformalization span domains that use specialized, low-resource DSLs: from SMT-LIB for constraint solving to Cedar for access control policies. Organizations frequently adopt niche or even create private DSLs tailored to their specific verification needs. While formal codebases may exist in isolation, these DSLs lack the large *paired* informal–formal corpora that Lean enjoys, making them difficult for data-driven approaches.

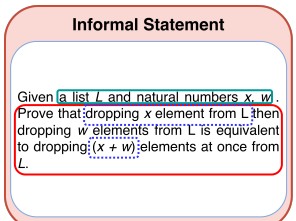 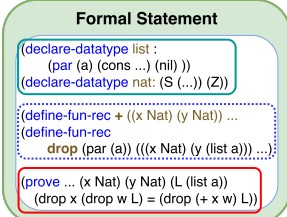

*Figure 3.* Informal-to-formal translation example (ATP). Formal statements are extracted from the Tons of Inductive Problems (Claessen et al., 2015) benchmark.

**a. Automated Theorem Prover (ATP) Languages.** Automated theorem provers operate over DSLs for encoding logical constraints. SMT solvers accept specifications in SMT-LIB (Barrett et al., 2010), while first-order theorem provers such as E Prover (Schulz, 2002) and Vampire (Riazanov & Voronkov, 1999) use TPTP syntax (Sutcliffe, 2024). These tools have achieved remarkable success in competitions such as SMT-COMP (Barrett et al., 2005) and CASC (Sutcliffe, 2016). Figure 3 illustrates formalizing an informal proof objective into SMT-LIB. The informal statement describes a property of the $drop(x, L)$ function: $\forall x, w, L. drop(w, drop(x, L)) = drop(x + w, L)$.

Autoformalization for this domain faces two key challenges. First, existing benchmarks (Claessen et al., 2015; Barrett et al., 2005; Sutcliffe, 2016) contain only formal specifications without natural language descriptions—for instance, the Tons of Inductive Problems benchmark (Claessen et al., 2015) provides the formal statements in Figure 3 but no informal counterpart. Second, informal statements may reference multiple theories (e.g., integers and lists in Figure 3) without explicit context, requiring theory-level formalization to identify necessary axioms and lemmas (*e.g.*, commutativity of addition). In general, the scarcity of aligned natural–formal specification pairs and theory-level formalization datasets remains an open challenge in the autoformalization of ATP languages.

**b. Protocol Verification Languages.** Verification of distributed protocols provides formal guarantees that a protocol design satisfies safety properties (*e.g.*, mutual exclusion), by proving these properties hold across all reachable states. State-of-the-art techniques take as input the protocol de-

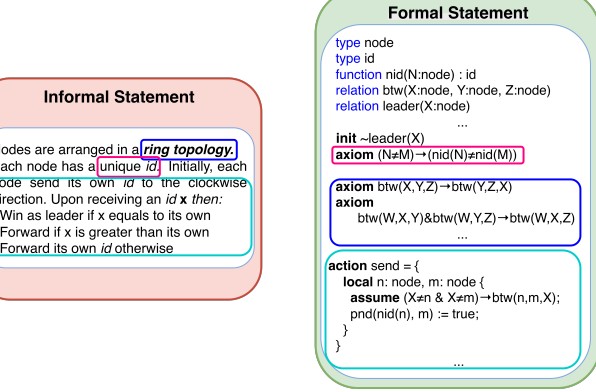

*Figure 4.* Informal-to-formal translation of a leader election protocol (Chang & Roberts, 1979).

sign written in domain-specific modeling languages such as Ivy (Padon et al., 2016) and PVerifier (Mora et al., 2023), along with inductive invariants that imply safety properties.

Autoformalization for this domain faces similar challenges. First, the domain is inherently low-resource: the number of widely-studied distributed protocols is small compared to mathematical theorems (Mathlib Community, 2020). The most comprehensive benchmark to date, IvyBench (Goel & Sakallah, 2020), contains *only 54 formalized proofs of protocols*. Second, formalization demands extensive domain knowledge about background theories. For example, Figure 4 shows an informal-to-formal translation of the Chang-Roberts leader election protocol (Chang & Roberts, 1979). In this example, the ring topology and initial states are encoded logically with axioms, and the formal model references atomic predicates introduced in this encoding. Theory-level datasets are needed to train LLMs on encoding such domain knowledge.

**c. Hardware Verification Languages.** Hardware security and correctness rely on formal verification to ensure designs behave as intended before fabrication. Engineers must translate informal design documents into formal verification languages such as SystemVerilog Assertions (SVAs), which express temporal properties over signal behaviors. This translation is labor-intensive, making it a natural candidate for autoformalization.

However, while datasets of hardware designs in Verilog exist (Thakur et al., 2024), designs with formal specifications remain scarce. Moreover, formalization is inherently challenging because design documents may combine natural language and timing diagrams, each containing ambiguity and implicit assumptions/assertions. For example, in Figure 5, the property stated informally as *control information from the source remains stable* implicitly asserts that it holds *starting from the next cycle* (captured by the ##1 operator in the SVA), which becomes apparent only when cross-referencing the timing diagram with the text.

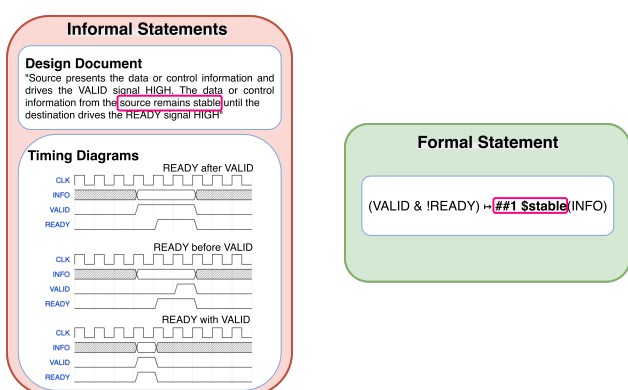

*Figure 5.* Informal-to-formal translation example from Shih et al.. The protocol is from the AMBA AXI specifications (Arm, 2004).

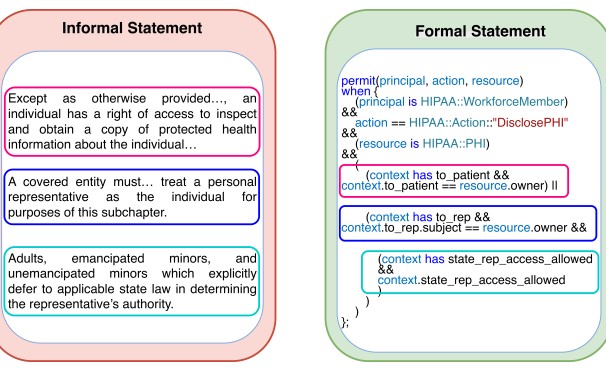

*Figure 6.* Informal-to-formal translation example of Cedar policy that implements clauses in HIPAA regulations.

Accurate formalization thus requires cross-modal reasoning to resolve inconsistencies and surface hidden assumptions. Developing autoformalization techniques capable of such multi-modal reasoning for articulating formal properties remains an essential yet largely unaddressed challenge.

**d. Declarative Programming Languages.** Declarative languages like SQL, Cypher, CodeQL, and Cedar let users specify *what* they want while delegating *how* to compute it to an execution engine. Unlike imperative languages, declarative program synthesis is a semantics-preserving translation from informal intent to formal, which is in the scope of autoformalization (see Appendix A for detailed discussions). We highlight 3 representative cases:

*Cypher*, the query language for graph databases like Neo4j, exemplifies low-resource DSL challenges. The Text2Cypher benchmark (Ozsoy et al., 2025) provides 44,387 natural language–query pairs, yet LLM performance remains modest: GPT-4o achieves only 33% exact match accuracy, improving marginally from 31% without fine-tuning. The difficulty stems from compositional semantics that require multi-hop reasoning over graph structures—patterns rarely seen in pretraining data and highly dependent on schema-specific knowledge that varies across deployments.

*CodeQL*, GitHub's query language for static security analysis, illustrates yet another dimension of difficulty: the application requires dual expertise in security vulnerabilities and program analysis. As shown by Wang et al. (2025a), synthesizing CodeQL queries from CVE descriptions is extremely challenging—Claude Code achieves only a 10% success rate when generating queries that correctly detect vulnerabilities in 176 CVEs across 111 Java projects. Even with sophisticated agentic scaffolding including language server feedback and retrieval-augmented generation, the success rate is only 53.4%.

*Cedar*, Amazon's policy language for authorization (Cutler et al., 2024), presents a different challenge: the informal inputs are enormous. Real-world policy documents like

HIPAA regulations (U.S. Department of Health and Human Services, 2024) span hundreds of pages of legal prose with nested conditions, exceptions, and cross-references. For example, Figure 6 shows how three separate HIPAA clauses about individual access rights, personal representatives, and minors must be jointly formalized into Cedar policies that correctly encode their interactions through shared predicates like `to_patient` and `state_rep_access_allowed`. While researchers have attempted to formalize fragments of such documents into Cedar, no complete large-scale policy corpus has been fully formalized. The gap between paragraph-level demonstrations and document-level formalization remains vast, requiring theory-level autoformalization that can maintain consistency across thousands of interdependent policy rules.

### 4.4. Challenge 4: Beyond Text - Real-World Applications Requiring Multimodal Inputs

Real-world formalization tasks often involve multimodal inputs. We identify four main modalities: (i) *Natural Language*, including descriptions and documentation; (ii) *Formal Languages*, such as mathematical notation or existing specifications; (iii) *Structured Diagrams*, including timing diagrams, schematics, and flowcharts; and (iv) *Free-Form Images*, such as geometric figures and UI mockups. These scenarios demand systems that reason across modalities, a capability current text-centric approaches lack.

In geometry, spatial relationships like betweenness and intersection are naturally conveyed by diagrams but cumbersome to express in text. Similarly, hardware timing diagrams and protocol state machines are primary carriers of formal content with implicit semantics (e.g., "starting from the next cycle") that must be parsed and integrated with textual descriptions through cross-modal reasoning. Finally, formalization sometimes requires translation *between* formal languages (e.g., Coq to Lean), which is non-trivial due to differing foundational assumptions and tactic ecosystems (Stoskopf et al., 2025), and motivates the common intermediate representation we propose in Section 5.3.

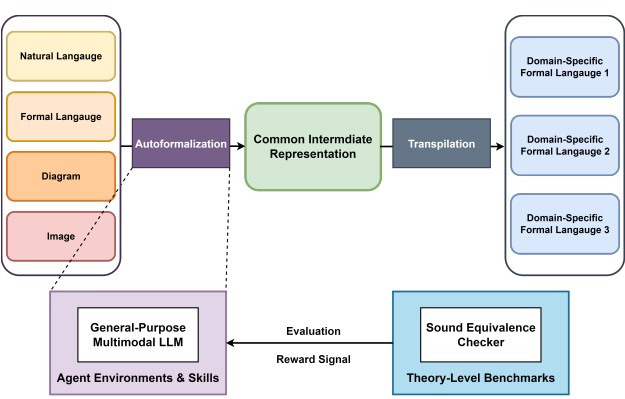

*Figure 7.* Our 3 proposals for theory-level autoformalization: a common intermediate representation, general-purpose LLM agents, and benchmarks with sound equivalence checking.

# 5. Call to Action: Proposals for Advancing Theory-Level Autoformalization

## 5.1. Proposal 1: Theory-Level Benchmarks with Sound Equivalence Checking

Existing benchmarks only evaluate definitions, statements, and proofs in isolation. We need benchmarks that evaluate entire theories as unified artifacts, including controlled tests of decomposition and abstraction learning (Section 4.2). Concretely, an ideal theory-level benchmark should satisfy 3 criteria: (i) it should include formalization of background theories, not just isolated statements; (ii) it should use sound equivalence checking with reported precision and recall compared against expert judgments (Section 4.1); and (iii) it should enforce data leakage prevention to ensure the background theories are not already in the model's training data.

We propose the following priority ordering for measuring progress. **Immediate:** theory-level benchmarks satisfying the above criteria. **Near-Term:** end-to-end case studies measuring the time and expert effort to formalize a new theory with AI assistance versus without; additionally, the community should maintain a tracking table of autoformalization performance across the DSLs surveyed in Section 4.3, so that progress in each domain can be monitored and compared over time. **Longer-Term:** evaluating whether a common IR-based approach (Section 5.3) offers genuine advantages over direct formalization.

## 5.2. Proposal 2: General-Purpose Models over Fine-Tuned Specialists

Specialized fine-tuned models suffer from severe overfitting: Goedel-Formalizer-V2-32B (Lin et al., 2025b) scores 0.0% on LeanEuclidPlus (a Lean benchmark outside Mathlib), while its base model Qwen3-32B achieves 20.6%—a pattern observed across all fine-tuned autoformalizers (Min et al., 2026). We therefore advocate building on general-purpose LLMs, enhanced through general training techniques like iterative SFT (Ying et al., 2025) and reinforcement learning with verifier feedback (Huang et al., 2025), combined with inference-time methods such as dependency retrieval (Zhang et al., 2025b; Wang et al., 2025b) and verifier feedback loops (Poiroux et al., 2025; Min et al., 2026). This approach transfers across DSLs and automatically benefits from foundation model improvements.

To be precise, this proposal is not about the training techniques themselves—SFT, RLVR, and test-time scaling are all standard—but about the *Data Mixture*, *Reward Design*, and *Evaluation Scope*. By "general-purpose," we mean models trained on diverse formal corpora spanning multiple domains and DSLs, with rewards that do not overfit to a single library's conventions, and evaluated on benchmarks including real-world textbooks and important proofs beyond standard competition mathematics.

## 5.3. Proposal 3: A Common Intermediate Representation for Theoretical Discovery

A common intermediate representation (IR) can address multiple challenges simultaneously. First, as depicted in Figure 7, the IR provides a unified target for formalizing multimodal inputs, enabling a complete and coherent specification even when the source materials are heterogeneous. Second, as discussed in Section 4.3, our target DSLs include low-resource, evolving, and proprietary languages; training LLMs to master each one individually is impractical. An IR reduces this combinatorial problem to a linear one. Third, a general IR combines the theorem-proving strengths of itself and the target DSLs: the DSLs often integrate tightly with specialized symbolic Automated Theorem Proving (ATP) tools, while the IR benefits from richer training data and thus LLM-automated Interactive Theorem Proving (ITP).

The design of such an IR is subject to 3 constraints: *Expressiveness* (represent diverse DSL semantics), *Verifiability* (type checking and proof verification within the IR), and *Embeddability* (deep embeddings of target DSLs for verified transpilation). Lean is a natural candidate: its dependent type theory provides expressiveness, its built-in type checker provides verifiability, and its metaprogramming facilities enable DSL-specific embeddings for verified transpilation within a single framework.

# 6. Conclusion

We have argued that autoformalization must move beyond isolated statements to formalize entire theories as holistic formal knowledge bases. We identified open challenges and proposed paths forward, aiming to turn autoformalization from a per-statement demonstration into a scalable tool for real-world verification and theoretical discovery.

## Acknowledgement

We would like to thank Noopur Bhatt, Yu-An Shih, Isil Dillig, and Ziyang Li for valuable discussions. We also thank the anonymous reviewers for their constructive feedback.

Marcus J. Min and Zixuan Yi are each supported by an AWS ASSET Ph.D. Fellowship. Mike He, Sharad Malik, and Aarti Gupta are supported by NSF Award CCF-2422053. Zhaoyu Li and Xujie Si are supported by the ProML project, funded by the Canadian Natural Sciences and Engineering Research Council and the French National Research Agency, under the reference ANR-25-CE23-6715. Osbert Bastani is supported by NSF Award CCF-2338777, Amazon Research Award Fall 2023, Amazon/ASSET Gift for Research in Trustworthy AI.

Any opinions, findings, conclusions, or recommendations expressed herein are those of the authors and do not necessarily reflect the views of funding entities.

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

## A. Declarative Program Synthesis vs. Imperative Program Synthesis

Autoformalization is, by definition, the semantics-preserving translation from informal natural language to a formal language: the informal input and the formal output should carry the same semantic content.

**Declarative Program Synthesis** satisfies this criterion. Declarative languages such as Prolog, SQL, Cedar, and CodeQL specify *what* should hold rather than *how* to compute it. Translating natural language into these languages encodes the information content of the informal input into a formally scoped specification, preserving its semantics. For example, translating "every nurse may read a patient's record only if assigned to that patient" into a Cedar policy produces a formal artifact whose meaning is precisely the informal statement; no implementation details are introduced.

**Imperative Program Synthesis**, by contrast, does not satisfy this criterion. Translating a natural-language task description into procedural or object-oriented code (e.g., Python, C++) introduces implementation details (control flow, data structures, algorithmic choices, security constraints, etc.) that are absent from the informal input. The output contains strictly more information than the input, making this a *generative* task rather than a semantics-preserving translation.

This distinction justifies our inclusion of declarative DSLs in Section 4.3 as autoformalization challenges. This view is shared by Mensfelt et al. (2025), who similarly consider declarative programming languages (Prolog, PDDL, OWL) as autoformalization tasks within a common framework. We advocate an inclusive view of autoformalization that recognizes these shared challenges and enables transfer of methods across domains: progress in mathematical autoformalization directly benefits declarative program synthesis, and vice versa.

## B. Proof Autoformalization vs. Formal Theorem Proving

Proof Autoformalization and Formal Theorem Proving have related but distinct goals:

**Formal Theorem Proving** aims to produce a formal proof that a given formal statement is true. The goal is to find *any* proof that passes the type checker or proof assistant, regardless of how it relates to existing informal arguments. Methods range from automated tactic search (Polu & Sutskever, 2020; Yang et al., 2023) to reinforcement learning over proof steps (Xin et al., 2024; Ren et al., 2025).

**Proof Autoformalization** aims to faithfully translate a *specific* informal proof into a formal one. The goal is not merely to verify that a statement is true, but to produce a formal proof whose logical structure corresponds to the reasoning in the original informal argument. This distinction is critical: two correct formal proofs of the same theorem may follow entirely different proof strategies, but only one of them may faithfully represent the informal proof being autoformalized.

Early work on proof autoformalization used the informal proof to guide formal proof construction: DSP (Jiang et al., 2023) translates the informal proof into a formal proof sketch of intermediate conjectures, then uses the sketch to guide an automated prover toward smaller subgoals; SPADeR (Tarrach et al., 2024) iteratively fills in missing implicit steps by attempting verification, identifying where the proof fails, and using an LLM to add the needed detail.

More recent work has focused explicitly on structural faithfulness: Step-Proof (Hu et al., 2025) decomposes a natural-language proof into sentence-level subproofs that can be verified individually; ProofFlow (Cabral et al., 2025) models the proof structure as a dependency DAG and formalizes each step as an intermediate lemma so the final Lean proof preserves the original logical flow; ProofBridge (Jana et al., 2025) uses a joint embedding model that aligns NL–FL theorem/proof pairs for cross-modal retrieval, combined with iterative proof repair via Lean feedback; and Chain of States (Wang et al., 2025c) extracts a sequence of intermediate formal proof states aligned to the informal reasoning steps, then generates tactics to move between adjacent states.

On the theory-level, it is not required to prove everything provable in a domain. The goal is to translate existing informal knowledge (axioms, definitions, notations, theorems, and their proofs) into a coherent formal library. If a true statement currently has no known informal proof, we would not expect autoformalization to produce a formal proof of it; that is the job of theorem proving. Theory-level autoformalization produces the formal foundation upon which theorem provers can then operate to discover genuinely new results.

