# OpenReview forum: "Position: Theory-Level Autoformalization, From Isolated Statements to Unified Formal Knowledge Bases"
_ICML.cc/2026/Position_Paper_Track — ICML 2026 Position Paper Track spotlight_

### Official Review · Reviewer_wGFs · 2026-02-14

**Significance:** 2
**Argument Clarity:** 3
**Rating:** 4
**Confidence:** 3

**Questions:**

1. Can you state, in one single paragraph of at most 5 sentences, what *exactly* the difference is between the theory-level formalization you propose and what the field is currently **aiming for** is (in your opinion)?
2. The link to abstraction learning is particularly interesting, but the discussion currently reads mostly like a literature review. Can you clarify what role abstraction learning would play in theory-level formalization, and what some of the alternatives might be? Does it matter if the learning is done symbolically (as in DreamCoder and follow up works like Bowers et al., POPL'23, Cao et al., POPL'23 or Grand et al., ICLR'24), or could abstraction learning through agent-driven summarization play the same role?
3. Under the strong definition of theory-level autoformalization as "automatically formalizing the entire theoretical context in a certain scope including axioms, definitions, notations, examples, lemmas, theorems, proofs, tactics, and the intertwined dependencies among them", it appears to be practically equivalent to "proving everything there is to prove in the given domain". Do you agree with this viewpoint? Why or why not?

**Alternative Views Section:**

Yes

**Compliance With Llm Reviewing Policy A Conservative:**

Affirmed.

**Discussion Potential:**

2

**Final Justification:**

This paper benefitted greatly from a productive and extensive discussion period. I thank the author for a particularly insightful discussion, and for pointing out my own misunderstanding regarding the discussion on declarative program synthesis.

My initial concerns were centered around:
- Writing: The paper spends a great deal of time belabouring certain points such as specific advances made within some branch of mathematics. While this does nicely motivate the problem space, it comes at the cost of the overall flow of the paper.
- Novelty/impact: Is the distinction being made between theory-level and existing approaches to autoformalization meaningful? Is it actionable? I am not convinced that the direction outlined herein does not already align with the community's vision for the field.

These concerns were **partially addressed** during the rebuttal period. The writing and narrative have improved somewhat, but the paper still lacks even a conclusion section. It is also clear from other reviewers' responses that there are members of this community who believe the position being argued for to be interesting and meaningful; however, having read and re-read the paper, I still have a hard time seeing it, and the arguments laid forth by authors themselves have not convinced me.

On balance, I therefore vote to marginally accept the paper. However, had the other reviewers not been so zealous, I would not have been sufficiently convinced by the paper itself to accept it; I therefore lower my confidence to 3.

**Paper Summary:**

This position paper advocates for a new research priority within the field autoformalization. It argues that the field currently focuses too much on what it calls "statement-level autoformalization" -- to what extent a model/agent/system can formalize individual statements in isolation. Instead, it advocates for a "theory-level" approach, where the emphasis is on the broader process of constructing and building corpora of knowledge from the ground up, starting with axioms, then reusable definitions, then lemmas, and then finally theorems. By doing so, it is argued, the field will develop techniques that are more well-suited for real-world formalization efforts across STEM. In addition, the paper discusses a few challenges that currently stand in the face of this progress (e.g., scaling subgoal decomposition) as well as some alternative viewpoints (importantly including a brief steelmaning of the view that statement-level autoformalization is sufficient). The paper concludes by outlining a few key research directions: constructing "theory-level" benchmarks with equivalence checking as a first-class property (e.g., so that syntactically distinct by semantically equivalent lemmas or definitions are equated); focusing on the use of general-purpose models instead of ones specialized to a particular domain; and developing a unified intermediate representation for autoformalization.

**Position:**

Yes

**Position In Title:**

Yes

**Related Work:**

4

**Strengths And Weaknesses:**

### Strengths
- The paper is well written, with only a single typo as far as this reviewer can see (a missing period on line 220; second column). Similarly, the figures are clear and help with the exposition of the position.
- The paper collects and references related work in a way that both advocates for the position and clarifies what current limitations there are with existing work.
- Autoformalization, and AI for mathematics more broadly, is a hot topic of likely interest to many attendees at ICML.
- The discussion of why sound equivalence checking is needed to construct theory-level benchmarks is illuminating and possibly of great interest to those working on evaluations in this field.
-  Proposal 3 (section 5.3), that the community should emphasis and transition to a unified intermediate representation for all formalization efforts, is very interesting. This could likely make for an entire position paper by itself, which could for example highlight the challenges -- such as the fact that even among ITPs their logical foundations differ greatly, which in turn affects which statements can be described easily and which become painfully laborious. If done well, this could spark some very interesting debate in the theorem proving community.

### Weaknesses
- The position is somewhat diffuse and poorly scoped. It is not clear how "theory-level autoformalization" is inherently different from the type of autoformalization that the literature is focused on today, other than collecting the results into reusable, shared knowledge bases.
- Similarly, the arguments used to advocate for (or illuminate challenges of) the position sometimes appear to go off-track. As one particularly egregious example, the section which lists "Declarative programming languages" as a challenging domain for autoformalization muddies the waters by conflating autoformalization and program synthesis. The same issue occurs in the paragraph around lines 428-430, which states that "[Translation between different formal languages] demand systems that reason across representational boundaires [...]", the relevance of which to the position at hand seems marginal at best.
- The fundamental position itself does not seem to be one that will spark much discussion or debate in the community, since it proposes a goal that (this reviewer believes) large sections of the community are already aligned with. While the introduction does start with the claim that "[a]utoformalization, translating informal natural language into formal, machine-verifiable languages, has been predominantly framed as a tool to generate training data for neural theorem provers, with most work focusing on individual statements", only the last part of this statement rings true to this reviewer's ears.
- The writing could be improved, particularly when it comes to the exposition of the position itself (rather than related work). There is no conclusion section whatsoever, and the (brief) introduction essentially just repeats the abstract in slightly more words. The paper may not need to go into quite so many details -- such as the exact nature of some particular advance in category theory -- to get its point across, and that would free up more space to clarify what exactly that point *is*.
- In short, the paper appears be trying to argue for too large and vague of a concept all at once. There are many interesting ideas in here, some of which could be turned into pointed position papers themselves, but they are lost in the crowd.

**Support:**

2

---

> ### Author Rebuttal · Authors · 2026-03-31
>
> Dear Reviewer wGFs,
>
> Thank you for your engaging and thoughtful review!
>
> ## Weaknesses
>
> ### W1: Position is diffuse; how does theory-level autoformalization differ from current efforts?
>
> We appreciate this question. Our key position is simple:
>
> 1. The community effort in autoformalization has been focused on statement-level translation, assuming that the formal theoretical context (axioms, definitions, notations, supporting lemmas, tactics) already exists.
> 2. We encourage the community to invest more into an entirely new problem: **theory-level autoformalization**, which requires fundamentally different capabilities that the field has not yet developed or evaluated.
>
> Why is theory-level a fundamentally different problem? To put it in one oversimplified but illustrative sentence:
>
> **Current research autoformalizes math statements and proofs based on Lean Mathlib; theory-level autoformalization aims to autoformalize Mathlib itself**.
>
> Concrete example: statement autoformalization on LeanEuclid can reach 71.4% ([DNA, Min et al. 2026](https://openreview.net/forum?id=NjgaeXNit3)), but this only evaluates Layer 3 (statements and proofs) in **Figure 2**. Layers 0-2 (axioms, primitive definitions, derivative definitions, notations, lemmas, tactics), which constitute the bulk of the formalization effort, are **not even attempted or evaluated by any existing work**.
>
>
> ### W2: Declarative programming languages are out of scope.
>
> Thank you for raising this point, as it allows us to clarify the commonly misunderstood difference between declarative and procedural program synthesis.
>
> Autoformalization, by definition, is the translation from informal input to a formal language. The key criterion is **whether the translation is semantics-preserving**: the informal input and the formal output should carry the same semantics.
>
> Declarative languages satisfy this criterion: translating natural language into Prolog, Cypher, or Cedar encodes *what* should hold, preserving the information content of the informal input. This is **fundamentally different from procedural program synthesis**, where the output contains implementation details (control flow, data structures) absent from the input, making it a generative task rather than a semantics-preserving translation.
>
> Therefore, **we advocate an open and inclusive view of autoformalization** that recognizes these shared challenges and enables transfer of methods across domains, because progress in mathematical autoformalization directly benefits these domains, and vice versa.
>
> This view is not unique to our paper. **[Towards a Common Framework for Autoformalization](https://arxiv.org/abs/2509.09810)** similarly considers declarative programming languages (Prolog, PDDL, OWL) as autoformalization tasks.
>
> ### W3: The position will not spark debate; the community is already aligned.
>
> We respectfully offer a different reading of the current state of the community consensus.
>
> First, **Reviewers dMMz, VwCK, and wGFs all recognized the alternative views in Section 3 as genuinely held.** As Reviewer dMMz noted, "these perspectives are indeed common in the community." Recent purely informal mathematical discoveries ([Erdos Problems](https://arxiv.org/abs/2601.22401), [Hirzebruch Proportionality](https://arxiv.org/abs/2601.23245) provide reasonable grounds to prioritize informal reasoning over formal. Our advocacy for theory-level is also far from consensus.
>
> Second, **theory-level is a completely new problem with many unexplored challenges and approaches.** The current state of the field confirms this:
>
> - **Benchmarks**: every existing benchmark evaluates statement-level translation only. No theory-level benchmark exists.
> - **Methods**: all published systems operate at the statement level. No system attempts end-to-end theory construction.
> - **Training**: the dominant approach is fine-tuning on Mathlib-style data, leading to severe overfitting (0.0% on DSLs like LeanEuclidPlus).
> - **Evaluation**: no sound equivalence checker with practical coverage exists (Section 4.1c).
>
> The absence of all of these suggests that this is a direction the community has not yet seriously pursued, which is exactly why we believe this position paper can spark productive discussion.
>
> ### W4: Writing can be improved.
>
> Thank you. We have revised the abstract and introduction. See the **[revised manuscript](https://github.com/anonymousauthor567/Theory-Level-Autoformalization/blob/main/_ICML_2026__Position__Autoformalization.pdf)**.
>
> ### W5: Paper tries to argue for too large a concept.
>
> The challenges and proposals are deeply interconnected: Proposal 1 addresses Challenges 1-2, Proposal 2 addresses Challenges 2-3, and Proposal 3 addresses Challenges 3-4. A narrower paper would miss the systemic nature of the problem.
>
> ## Thank You Note
>
> Due to space constraints (5000 chars), we will address the 3 great questions you raised in the discussion phase. Thanks again for your time!

---

> > ### Author Rebuttal · Reviewer_wGFs · 2026-04-02
> >
> > I thank the authors for their very well structured and worded reply to my comments.
> >
> > > W2
> >
> > This was completely my oversight, and I thank the authors for correcting me. *Declarative* program synthesis indeed seems like a well-justified example of a high-impact, real-world autoformalization task.
> >
> > > W1 and W3
> >
> > Although I do not agree with the other reviewers that such theory-level autoformalization is not already a *goal* of the community (I was not referring to the other alternative viewpoints, such as the supposed sufficiency of informal reasoning that the authors mention in their reply), I certainly agree with the analysis that this is at least not *acted upon* by the community at this time.
> >
> > My experience with the autoformalization community is that it is similar to where program synthesis was at 2-3 years ago. At the time, code generation papers were still stuck running evals on MBPP, APPS and HumanEval, but the community still dreamt of producing systems that could generate an entire piece of software from scratch; hill-climbing on competitive programming benchmarks was just a way of slowly building up the knowledge and mechanisms needed to get there. Similarly, I have certainly never interacted with someone in this community who did not dream of achieving what you call theory-level autoformalization, even if they were currently occupied with hill-climbing on some benchmark.
> >
> > A more charitable reading of this paper's position is perhaps that the authors believe **now** is the time for the community to take on this challenge, regardless of whether it did or did not already believe it to be a goal, as we have made sufficient progress elsewhere and are ready to take on a bigger job. This is a position that I think offers far more potential for interesting discussion, and perhaps with some rewording that message could make its way into the manuscript for the benefit of those who already have ambitious goals for their research?
> >
> > > Thank you note
> >
> > I look forward to hearing the author's thoughts on questions (2) and (3) from my original review (the response to W1 already answered question 1).

---

### Official Review · Reviewer_VwCK · 2026-03-12

**Significance:** 4
**Argument Clarity:** 3
**Rating:** 5
**Confidence:** 5

**Questions:**

1. The paper’s central claim is that statement-level autoformalization is too narrow. Could the authors provide one or two concrete examples where a system that succeeds at statement-level translation would still fail at theory-level autoformalization, and explain which missing capabilities are essential in those cases?

2. The paper advocates for theory-level benchmarks with sound equivalence checking. What, concretely, would the authors require for such a benchmark to be considered valid and useful in practice? In particular, what artifacts, supervision, and evaluation criteria are essential, and which are optional?

3. A key proposal is to develop a common intermediate representation across domains and languages. Could the authors clarify what design constraints such an intermediate representation must satisfy, and what tradeoffs they see between expressivity, semantic faithfulness, interoperability, and practical adoption?

4. The paper argues that general-purpose models are preferable to narrow fine-tuned specialists. Could the authors clarify whether this is meant as a near-term practical recommendation, a long-term research preference, or both? What kind of evidence would lead them to revise this view?

5. If the community were to follow the paper’s recommendations, what would the most convincing near-term demonstration of progress look like? For example, would the authors view a new benchmark, a cross-domain system result, a theory-construction case study, or an interoperability framework as the most meaningful first milestone?

**Alternative Views Section:**

Yes

**Compliance With Llm Reviewing Policy A Conservative:**

Affirmed.

**Discussion Potential:**

3

**Final Justification:**

After reading the paper and the rebuttal, I remain positive on the submission. I thought the core position was clear and useful, and the main strengths were the clarity of the argument, the practical artifacts, and the relevance of the topic to benchmark design and reproducibility. My earlier concerns were mostly about scope, how broadly the claims should generalize beyond the main case study, and the practical impact of the proposed artifacts.

The rebuttal helped on those points. In particular, it made the distinction between core requirements and stronger best practices much clearer, and it gave a more concrete sense of the practical stakes!

**Paper Summary:**

I found this paper to be arguing that the field should move beyond statement-level autoformalization and place more emphasis on **theory-level autoformalization**. The authors seek to discuss a central concept: that real formalization work is rarely about translating a single theorem in isolation, and instead usually requires building a much larger formal structure of definitions, lemmas, proofs, tactics, and dependencies. The paper argues that this broader view is important because many meaningful formalization efforts in mathematics, science, and engineering are naturally theory-level rather than statement-level.

Overall, the authors examine an important theme: what kind of autoformalization agenda would actually scale to realistic verification and scientific discovery. I found that the paper not only argues for this shift in emphasis, but also tries to support it by addressing alternative views, identifying major open challenges, and proposing future directions such as theory-level benchmarks, stronger equivalence checking, and shared intermediate representations across languages and domains.

**Position:**

Yes

**Position In Title:**

Yes

**Related Work:**

3

**Strengths And Weaknesses:**

I thought the paper’s main strength was the clarity of its central position. The authors are not merely saying that autoformalization matters; they make the more specific argument that the field should prioritize **theory-level autoformalization** rather than focusing too narrowly on isolated statement translation. I found this position easy to understand and well motivated, especially because the paper repeatedly connects it to how formalization actually happens in practice: through the construction of definitions, lemmas, abstractions, proof infrastructure, and larger libraries rather than one-off theorem conversion. That makes the paper relevant not only to theorem proving, but also to broader ICML discussions about how research agendas are shaped by benchmarks and task formulation.

I also found the paper strong in the way it opens up a broader research agenda rather than presenting a narrow complaint about current benchmarks. The discussion of alternative views is useful, and I appreciated that the paper tries to engage with credible competing perspectives instead of ignoring them. The manuscript also points to concrete future directions, including theory-level benchmarks, equivalence checking, decomposition and abstraction, multimodal inputs, and intermediate representations across domains and theorem provers. That gives the position practical substance and makes the paper likely to generate discussion within the ICML community, especially among researchers interested in reasoning, formal verification, and domain-specific scientific applications.

My main weakness is that, while the position is clear and intuitive, the support is more conceptual than empirical. The paper does a good job explaining why theory-level autoformalization is important, but I found less concrete evidence showing how much is lost by current statement-level approaches or how progress should actually be measured under the proposed shift. In other words, the normative case is strong, but the evidentiary case is thinner. For a position paper that is trying to redirect a research agenda, I would have liked to see a sharper account of what existing evaluations systematically miss, perhaps through more detailed case studies, failure analyses, or comparisons between statement-level and theory-level outcomes.

I also thought some of the recommendations remain somewhat high level. The paper argues for theory-level benchmarks and shared intermediate representations, but I came away wanting a more precise sense of what the minimum actionable standard would be for the community. For example, what would count as a convincing theory-level benchmark, how should equivalence be validated in practice, and how should one balance general-purpose systems against more specialized tools in realistic domains? I think the paper cites relevant themes and related directions well enough for a position piece, but it would be more persuasive with a more operational roadmap and a clearer sense of which proposals are immediate priorities versus longer-term aspirations.

**Support:**

3

---

> ### Author Rebuttal · Authors · 2026-03-31
>
> Dear Reviewer VwCK,
>
> Thank you for your careful and constructive review!
>
> ## Weaknesses
>
> ### W1: Support is more conceptual than empirical.
>
> We fully agree with the importance of stronger empirical grounding. Since theory-level autoformalization is an entirely new direction, no theory-level benchmarks currently exist, and thus direct comparisons between statement-level and theory-level are not available at this point. However, we did **try our very best to collect both quantitative and qualitative evidence in Section 4** to show why current approaches, which are rooted in the statement-level view, are insufficient for real-world (mostly theory-level) applications.
>
> In Section 4.1, we show that existing statement autoformalization benchmarks have high error rates (31.8\% in ProofNet, 8.6\% in PutnamBench), showing that even statement-level evaluation is unreliable, let alone theory-level.
>
> In Section 4.2, we discuss how recent human-AI collaborations still require human-written blueprints, showing that the decomposition and abstraction capabilities needed for theory-level work are still missing.
>
> In Section 4.3, we document concrete failures in low-resource DSLs: Goedel-Formalizer-V2-32B scores 0.0\% on LeanEuclidPlus, GPT-4o achieves only 33\% exact match on Text2Cypher, and Claude Code achieves only 10\% on CodeQL generation.
> We also provide qualitative examples for each DSL (Figures 3, 4, 5) to illustrate the specific challenges.
>
> These are not conceptual arguments but quantitative and qualitative evidence that current methods fail precisely where theory-level capabilities are needed.
>
> ### W2: Need roadmap to measure progress towards theory-level autoformalization
>
> We propose the following priority ordering:
>
> 1. **Immediate: theory-level benchmarks.** Including background theory formalization (not just isolated statements), sound equivalence checking with reported precision/recall against expert judgments, and data leakage controls.
>
> 2. **Near-term: end-to-end case studies and cross-DSL tracking.** Measuring time and expert effort to formalize a new theory with vs. without AI assistance. The community should also maintain a tracking table of autoformalization performance across the DSLs in Section 4.3 (SMT-LIB, TPTP, Ivy, SVA, Cedar, Cypher, CodeQL).
>
> 3. **Longer-term: common IR and cross-domain generalization.** This requires mature theory-level benchmarks across multiple DSLs first.
>
> ## Questions
>
> ### Q1: Concrete example where statement-level success fails at theory-level.
>
> Consider [LeanEuclid](https://arxiv.org/abs/2405.17216). With proper decomposition and abstraction learning, statement autoformalization on LeanEuclid can reach 71.4% ([DNA, Min et al. 2026](https://openreview.net/forum?id=NjgaeXNit3)). However, this only evaluates Layer 3 (statements and proofs) in Figure 2 of our paper. Layers 0-2 (axioms, definitions, notations, lemmas, tactics) are built on [SystemE](https://arxiv.org/abs/0810.4315), which must be formalized from scratch since none of it exists in Mathlib. These lower layers are **not even attempted or evaluated** by any current method, yet they constitute the bulk of the formalization effort.
>
> ### Q2: What makes a good theory-level benchmark?
>
> Essential: (i) data leakage controls targeting domains outside standard libraries, (ii) sound equivalence checker with reported precision/recall against expert judgments (prioritizing soundness over completeness), and (iii) evaluation of the evaluation itself, comparing automated judgments to expert annotations.
>
> ### Q3: Design constraints for a common IR.
>
> Three constraints: *expressiveness* (represent diverse DSL semantics), *verifiability* (type checking and proof verification within the IR), and *embeddability* (deep embeddings of target DSLs for verified transpilation). Lean strikes a reasonable balance with dependent type theory, a built-in type checker, and metaprogramming facilities for DSL-specific embeddings.
>
> ### Q4: General-purpose models: near-term, long-term, or both?
>
> Both. Fine-tuned specialists already demonstrably overfit (Section 5.2), and theory-level autoformalization inherently requires generalization across diverse DSLs and domain theories. We would revise this view if a composition of specialized models consistently outperformed general-purpose models on theory-level benchmarks, but such evidence does not exist.
>
> ### Q5: Most convincing near-term demonstration.
>
> A **theory-construction case study**: end-to-end autoformalization of an informal source (e.g., a textbook chapter outside Mathlib) into a coherent, type-checking formal library. The recent [formalization of Munkres' Topology in Megalodon](https://arxiv.org/abs/2601.03298) illustrates the target; replicating this with significant AI assistance would be a compelling milestone.
>
> ## Thank You Note
>
> Please let us know if the above clarification resolves your concerns. We'd be more than pleased to elaborate more. Thanks again for your time and consideration!

---

> > ### Author Rebuttal · Reviewer_VwCK · 2026-04-02
> >
> > Thanks for the answers. It addressed my main concerns:)!

---

### Official Review · Reviewer_dMMz · 2026-03-14

**Significance:** 3
**Argument Clarity:** 3
**Rating:** 5
**Confidence:** 4

**Questions:**

- In 5.3, the authors claim that Lean is a natural candidate to be the common intermediate presentation that can potentially handle multimodal inputs. Could the authors elaborate on how this could work?
- Line 254, 'unrestricted propositional equivalence is not sound': I am still not quite sure where the unsoundness comes from, even if every step is in a formal environment. Additionally, I'm confused by the equivalence between 'n * 1 = n' and 'n + 0 = n'.
- In the abstract, you mention 5 challenges, but the main text only discusses 4. Is this a typo?

**Alternative Views Section:**

Yes

**Compliance With Llm Reviewing Policy A Conservative:**

Affirmed.

**Discussion Potential:**

3

**Paper Summary:**

The paper positions the need for theory-level autoformalization. In particular, it addresses 3 alternative views (informal over formal reasoning, theorem proving over autofformalization, statement autofformalization over theory-level autofformalization). It also clearly identifies 4 current challenges (for theory-level autofformalization): equivalence checking, hierarchical decomposition and abstraction learning, real-world applications using low-resource DSLs, and multimodal inputs. Finally, 3 proposals are laid down: theory-level benchmarks, aiming for general-purpose models, and a common intermediate representation language for handling multi-modal and ATP tools.

**Position:**

Yes

**Position In Title:**

Yes

**Related Work:**

3

**Strengths And Weaknesses:**

Pros
- Comprehensive review of existing work and clear identification of the problems
- Well-addressed alternative views: these perspectives are indeed common in the community, and I appreciate the solid counterargument
- Illustrative examples make this paper easy to follow

Cons
- Proposals for next steps are somewhat thin, and it remains unclear how the proposal addresses the identified challenges in theory-level autoformalization. For example, when proposing general-purpose models over fine-tuned specialists, the authors mention several techniques like SFT, RL with verifier feedback, and inference-time methods—all of which are standard when training specialized fine-tuned models. It's not obvious how combining these techniques yields general-purpose models.
- We lack a clear way to measure progress toward theory-level autoformalization: there are no identified milestones or benchmarks yet (though these need to be created).

**Support:**

3

---

> ### Author Rebuttal · Authors · 2026-03-31
>
> Dear Reviewer dMMz,
>
> Thank you for your thoughtful review and for supporting the importance of theory-level autoformalization! We appreciate your constructive suggestions on strengthening the proposals and clarifying technical details.
>
> ## Weaknesses
>
> ### W1: Proposals can be made more concrete; unclear how standard techniques (e.g. SFT, RLVR) yield general-purpose models.
>
> We agree that the individual techniques (SFT, RL with verifier feedback, inference-time methods) are standard. Our key argument is not about the techniques themselves but about **the data mixture, the RL reward design, and the evaluation scope**.
>
> Take SFT for example, specialized fine-tuned models like Goedel-Formalizer-V2-32B are trained predominantly on Mathlib-style data and evaluated on Mathlib-dependent benchmarks, which leads to severe overfitting: Goedel-Formalizer-V2-32B scores 0.0% on LeanEuclidPlus (a benchmark outside Mathlib), while its base model Qwen3-32B achieves 20.6%. By "general-purpose," we mean models trained on diverse formal corpora spanning multiple domains and DSLs, and evaluated on a wide set of benchmarks including real-world textbooks and important proofs beyond standard competition mathematics.
>
> ### W2: Need clear ways to measure progress towards theory-level autoformalization.
>
> This is an important concern. We agree that a clear roadmap to measure progress towards theory-level autoformalization is essential. We propose the following priority ordering:
>
> 1. **Immediate priority: theory-level benchmarks.** A valid benchmark should include (i) formalization of background theories (not just isolated statements), (ii) sound equivalence checking with reported false positive and false negative rates compared against expert judgments, and (iii) controlled data leakage prevention to ensure the background theories are not already in the model's training data.
>
> 2. **Near-term priority: end-to-end case studies and cross-DSL tracking.** Measuring the time and expert effort to formalize a new theory with AI assistance versus without, following the methodology of recent collaborations like the strong prime number theorem formalization. In addition, the community should maintain a tracking table of LLM-based autoformalization performance across the DSL applications surveyed in Section 4.3 (e.g., SMT-LIB, TPTP, Ivy, SVA, Cedar, Cypher, CodeQL), so that progress in each domain can be monitored and compared over time.
>
> 3. **Longer-term aspiration: common intermediate representation and cross-domain generalization.** This requires mature theory-level benchmarks across multiple DSLs to evaluate whether an IR-based approach offers genuine advantages over direct formalization.
>
> ## Questions
>
> ### Q1: How can Lean serve as a common IR for multimodal inputs?
>
> Lean is a natural candidate because it has the largest formal corpus (including synthesized data) and expressive metaprogramming facilities that allow deep embeddings of other DSLs.
>
> For multimodal inputs such as timing diagrams in hardware verification, a multimodal LLM can first interpret the visual input and extract the intended signal sequences, which are then formalized in Lean as concrete test cases for the target SVA properties.
>
> This would, of course, first require a Lean implementation of the SVA language, but it is a feasible task due to Lean's expressiveness and metaprogramming infrastructure. The Lean formalization thus serves as a unified representation for both input modalities.
>
> ### Q2: Why is unrestricted propositional equivalence unsound? Why are `n * 1 = n` and `n + 0 = n` equivalent under it?
>
> The issue is that if we allow arbitrary lemmas and theorems from the background theory to establish equivalence, then any two true statements in the background theory become equivalent.
>
> Concretely, say our background theory is Peano arithmetic, both `n * 1 = n` and `n + 0 = n` are provably true, so the (unrestricted) propositional equivalence `n * 1 = n ↔ n + 0 = n` can be proved. But these are clearly different mathematical claims: one is about the multiplicative identity and the other about the additive identity. An equivalence checker that accepts this would judge any true formalization as equivalent to any other true formalization, rendering evaluation meaningless. This is why we argue that the background theory must be restricted to an admissible fragment.
>
> ### Q3: "5 challenges" vs. 4 in the main text.
>
> Thank you for catching the typo! We have corrected it in our revised manuscript.
> Since **new submission of revised manuscripts are not enabled** at this point, we have uploaded the revised manuscript at this **[anonymous link](https://github.com/anonymousauthor567/Theory-Level-Autoformalization/blob/main/_ICML_2026__Position__Autoformalization.pdf)** for your reference.
>
> ## Thank You Note
>
> Please let us know if the above clarification resolves your concerns. We'd be more than pleased to elaborate more. Thanks again for your time and consideration!

---

> > ### Author Rebuttal · Reviewer_dMMz · 2026-04-02
> >
> > Thanks for the answers. All my concerns have been resolved. :-)

---

### Official Review · Reviewer_cQ3V · 2026-03-23

**Significance:** 4
**Argument Clarity:** 3
**Rating:** 5
**Confidence:** 4

**Questions:**

I have identified the following important weaknesses. Can you fix them relatively easily?

- The abstract over-emphasizes enumeration than conceptual clarity. For instance, the abstract currently foregrounds numbered lists ("3 alternative views," "5 open challenges," "3 promising paths") instead of first clearly conveying the conceptual novelty. It feels like a checklist-driven approach rather than argument-driven. I would suggest the authors to write these numbers in words and to prioritize the conceptual distinction earlier, which would improve readability.

- The introduction section also has a major structural weakness. The formal definition of theory-level autoformalization appears only in Section 2.2, even though the concept is central to the title, abstract, and introduction. Because of this, the abstract initially reads somewhat vague: the reader encounters claims about "3 alternative views," "5 open challenges," "3 promising paths" before fully understanding what differentiates theory-level autoformalization from statement-level work. This makes the opening far less effective than it could be. It delays the emergence of the actual position. The examples showing that manual formalization can take years or even decades (e.g., the Kepler Conjecture), while AI-assisted formalization can take much less time (e.g., the strong prime number theorem formalized in 1.5 weeks), are very compelling and should appear earlier. Likewise, the concepts of grounding and guiding, introduced in Section 2.1c, would be more impactful if presented sooner.

**Alternative Views Section:**

Yes

**Compliance With Llm Reviewing Policy A Conservative:**

Affirmed.

**Discussion Potential:**

3

**Paper Summary:**

This paper argues that autoformalization should move beyond isolated statement translation toward theory-level autoformalization, defined as constructing complete formal theories including axioms, definitions, lemmas, proofs, tactics, and dependencies. The authors seek to discuss a central concept: that meaningful progress in formal reasoning and machine-assisted theorem proving ultimately requires formalizing entire theoretical ecosystems rather than individual statements. Overall, the authors examine an important theme at the intersection of formal methods, theorem proving, and LLM-based reasoning, and they support their argument through examples from mathematics, software verification, hardware verification, and domain-specific languages.

**Position:**

Yes

**Position In Title:**

Yes

**Related Work:**

3

**Strengths And Weaknesses:**

Strengths
--------------
- The paper clearly states a position that the next major step for autoformalization research should be theory-level formalization rather than statement-level benchmarks. This aligns with the position-paper track because it explicitly advocates a research direction and explains why current trends may be insufficient.

- A major strength is the broad motivation across multiple domains. The discussion of large formalization efforts such as CompCert, Liquid Tensor Experiment, and the formal proof of the Four Color Theorem effectively demonstrates that real formalization is already theory-level in practice, and that human effort currently dominates the cost.

- Section 3 which discusses alternative views is one of the strongest parts of the paper. The authors engage with credible competing perspectives e.g., informal reasoning may suffice, theorem proving may be more important than autoformalization, statement-level formalization may already scale. The corresponding counterarguments presented are generally thoughtful, especially the observation that theorem provers presuppose formally specified goals and therefore depend fundamentally on formalization pipelines.

- The paper directly addresses an active fault line in current research: whether scaling theorem proving should focus on proving harder statements or building richer formal libraries. The context is stronger than many position papers because the argument is grounded in concrete prior work (covering recent benchmark papers, theorem proving systems, verification DSLs, and recent human-AI formalization efforts) rather than abstract opinion alone.

- The final section provides plausible call-to-actions e.g., theory-level benchmarks, sound equivalence checking, general-purpose models, common intermediate representations.

Weaknesses
--------------
- One of the most concerning weaknesses of this paper is in the abstract and introduction sections.

- The abstract over-emphasizes enumeration than conceptual clarity. For instance, the abstract currently foregrounds numbered lists ("3 alternative views," "5 open challenges," "3 promising paths") instead of first clearly conveying the conceptual novelty. It feels like a checklist-driven approach rather than argument-driven. I would suggest the authors to write these numbers in words and to prioritize the conceptual distinction earlier, which would improve readability.

- The introduction section also has a major structural weakness. The formal definition of theory-level autoformalization appears only in Section 2.2, even though the concept is central to the title, abstract, and introduction. Because of this, the abstract initially reads somewhat vague: the reader encounters claims about "3 alternative views," "5 open challenges," "3 promising paths" before fully understanding what differentiates theory-level autoformalization from statement-level work. This makes the opening far less effective than it could be. It delays the emergence of the actual position. The examples showing that manual formalization can take years or even decades (e.g., the Kepler Conjecture), while AI-assisted formalization can take much less time (e.g., the strong prime number theorem formalized in 1.5 weeks), are very compelling and should appear earlier. Likewise, the concepts of grounding and guiding, introduced in Section 2.1c, would be more impactful if presented sooner.

- The introduction provides very little detail and fails to convince the reader of what the rest of the paper contains (which is impressive in my opinion). Furthermore, the abstract and introduction repeat the same sentence verbatim about autoformalization being viewed mainly as data synthesis for theorem provers, creating redundancy early in the paper. It feels as though the abstract and introduction were written in a hurry and would benefit from a careful rewrite.

- The abstract and introduction begin with the statement that "auto-formalization...has been framed as a tool to generate training data for neural theorem provers,"" but this framing is inaccurate and unnecessarily narrow. Autoformalization is fundamentally motivated by the goal of verifying NL reasoning and proofs, not merely by data generation for theorem provers. The authors themselves acknowledge this in Section 3.2, where they explicitly present this framing as a position to be challenged. Thus, using it as the opening premise of both the abstract and introduction is conceptually inconsistent: if the paper later argues against this characterization, it should not be presented at the outset as the central framing of the paper.

- The text in Figure 1 is not legible; the authors should increase the font size to improve readability. Figure 6 is also difficult to follow, as it contains too many arrows and does not clearly convey what they represent.

Overall Recommendation
--------------
This is a strong and timely position paper that raises an important research direction for autoformalization: moving from isolated statement translation toward theory-level formalization. The paper's main argument is well supported through concrete examples across mathematics, verification, and theorem proving, and Section 3 is particularly effective in engaging with alternative perspectives. However, its main weakness lies in the abstract and introduction, which do not adequately reflect the strength of the later sections. The core concept is introduced too late, the opening relies too heavily on enumeration, and some framing is repetitive and conceptually inconsistent with later arguments. Overall, I view the paper positively, but the opening sections would benefit from substantial revision to match the clarity and strength of the paper’s central position.

**Support:**

3

---

> ### Author Rebuttal · Authors · 2026-03-31
>
> Dear Reviewer cQ3V,
>
>
> Thank you so much for your supportive assessment and constructive feedback on our work!
>
> ## Weaknesses & Questions
>
> Your feedback on the writing of the abstract and the introduction is very helpful. We have updated the manuscript to **emphasize the conceptual distinctions** between theory-level autoformalization and statement-level autoformalization over enumeration of the content.
>
> Thanks so much for pointing out the readability issues with the figures. We have increased the font size in Figure 1 and simplified Figure 6 to make them more readable.
>
> Since **new submission of revised manuscripts are not enabled** at this point, we have uploaded the revised manuscript at this **[anonymous link](https://github.com/anonymousauthor567/Theory-Level-Autoformalization/blob/main/_ICML_2026__Position__Autoformalization.pdf)** for your reference.
>
>
> > Autoformalization, translating informal natural language into formal, machine-verifiable languages, has been predominantly framed as a tool to generate training data for neural theorem provers, with most work focusing on individual statements.
> This position paper argues for theory-level autoformalization: formalizing complete theories, including axioms, definitions, notations, examples, lemmas, theorems, proofs, tactics, and all their inter-dependencies as entire formal libraries.
>
>
> Regarding our original opening sentence (quoted above), we apologize for the confusion. What we want to convey is that over the past few years, most work on autoformalization has focused on leveraging it as a data synthesis tool for neural theorem provers, with most work focusing on individual statements.
>
> However, as you correctly pointed out, this framing is exactly what our paper wants to argue against, by presenting a comprehensive view of motivations for autoformalization and a new problem with fundamentally new challenges: theory-level autoformalization, hoping to encourage the community to invest more effort into these new directions.
>
> ## Thank You Note
> Please let us know if the above revisions and clarifications resolve your concerns. We'd be more than pleased to elaborate more. Thanks again for your time and consideration!

---

### Decision · Program_Chairs · 2026-04-30

**Decision:**

Accept (spotlight)

**Comment:**

Reviewers are enthusiastic about this paper (to the extent that thoughtful academics can be enthusiastic :-)

The central position advocated is that automated formalization of mathematics should target complete theories including definitions and axioms, not just isolated lemmas and theorems. Here is an additional justification for this position: complete theories evolve over time as counter-examples and proof weaknesses are discovered for theorems that are true "in essence." This phenomenon is explained in detail in the book Proofs and Refutations by Imre Lakatos (1976). So a formalization that addresses just one lemma or theorem does not capture the true nature of mathematics, where definitions and proofs are established in a dialectic with each other.